# Storm Surge Inundation Analysis with Consideration of Building Shape and Layout at Ise Bay by Maximum Potential Typhoon

**Masaki Nimura [1,\*], Shuzo Nishida [2], Koji Kawasaki [1,2,3,4], Tomokazu Murakami [5] and Shinya Shimokawa [5]**

1   Hydro Technology Institute, Co., Ltd., Nakanoshima, Osaka 530-6126, Japan; kawasaki@hydrosoken.co.jp
2   Department of Civil Engineering, Osaka University, Osaka 565-0871, Japan; nishida@civil.eng.osaka-u.ac.jp
3   Department of Urban Design and Engineering, Osaka City University, Osaka 558-8585, Japan
4   Department of Civil Engineering, Meijo University, Nagoya 468-8502, Japan
5   National Research Institute for Earth Science and Disaster Resilience, Tsukuba 305-0006, Japan; tmurakami@bosai.go.jp (T.M.); simokawa@bosai.go.jp (S.S.)
\*   Correspondence: nimuramk@hydrosoken.co.jp

**Abstract:** Global warming is feared to cause sea-level rise and intensification of typhoons, and these changes will lead to an increase in storm surge levels. For that reason, it is essential to predict the inundation areas for the maximum potential typhoon and evaluate the disaster mitigation effect of seawalls. In this study, we analyzed storm surge inundation of the inner part of Ise Bay (coast of Aichi and Mie Prefecture, Japan) due to the maximum potential typhoon in the future climate with global warming. In the analysis, a high-resolution topographical model was constructed considering buildings' shape and arrangement and investigated the inundation process inside the seawall in detail. The results showed that buildings strongly influence the storm surge inundation process inside the seawall, and a high-velocity current is generated in some areas. It is also found that closing the seawall door delays the inundation inside the seawall, but the evacuation after inundation is more difficult under the seawall doors closed condition than opened condition when the high tide level exceeds the seawall.

**Keywords:** storm surge; three-dimensional numerical analysis; maximum potential typhoon; Ise Bay

## 1. Introduction

The storm surge's main causes are the pressure-driven effect due to the low atmospheric pressure and the wind-driven effect due to the strong wind in coastal areas. In the coastal areas of Japan, the impact of the wind-driven surge is significant. The wind-driven surge is exceptionally high in bays where the angle of the mouth of the bay coincides with the wind direction and the shallow water depth. A typical example of such a bay is Ise Bay, where a maximum storm surge deviation of 3.55 m by Typhoon Vera [1] was recorded. A storm surge deviation means the difference between the observed and the astronomical tide level, which are 3.89 m and 0.34 m by Typhoon Vera. The Ise Bay coastal area is a densely populated and large industrial area and has important facilities, such as the Nagoya Port and CHUBU CENTRAIR International Airport. Therefore, tidal protection facilities have been developed to protect the area from the maximum tide height of Typhoon Vera.

However, rising sea temperature due to global warming is feared by the intensification of typhoons and higher storm surges in the future [2]. If a storm surge more significant than that of Typhoon Vera occurs, large-scale inundation may occur inside the seawall. In response to the inundation, it is vital to

consider soft measures, such as hazard maps and evacuation plans, and detailed information on the inundation process inside the seawall is useful as basic information in the study of countermeasures.

In previous studies, there have been many studies of a two-dimensional planar model [3–5] or a three-dimensional hydrostatic model based on the σ-coordinate system [6,7] to analyze the inundation area due to storm surge. In these models, the building is generally represented indirectly as surface roughness. However, in urban areas where buildings are densely packed, the effect of buildings' shape and layout on the inundation process is significant. In recent years, to investigate the flow process of tsunami around structures in detail, multiple studies have been reported that consider the shape and layout of structures [8,9]. On the other hand, few studies consider the shape and layout of structures about storm surge inundation analysis [10].

In this study, we constructed a high-resolution geometry model considering the shape and layout of buildings around the Nagoya Port area in the inner part of Ise Bay. Then, we carried out a storm surge inundation analysis by the maximum potential typhoon due to global warming and investigated the effect of the shape and layout of buildings on the inundation process. In addition, we studied the impact of seawall doors conditions on inundation of storm surge.

## 2. Materials and Methods

### 2.1. Target Area

In this study, we mainly focused on the inner part of Ise Bay, which is facing Aichi and Mie prefectures in Japan. The coastal area of Ise Bay is a large city with a population of millions of people. The area is also one of leading industrial clusters in Japan, and the main industries are automotive and electronics. The Nagoya Port, located in the inner part of Ise Bay, is the largest port in terms of cargo handling volume in Japan.

The inner part of Ise Bay was severely inundated by the storm surge caused by Typhoon Vera (1959), killing 5089 people, which is the largest number of people in the storm surge disaster in Japan [1]. The storm surge overtopped the dikes, and some of the levees were breached. After Typhoon Vera, dikes and seawalls have been built to protect against the storm surge equivalent to that caused by Typhoon Vera. The probability of Typhoon Vera is estimated to be 100–150 years [11]. In this area, there are many studies that have examined the impact of global warming on storm surges [12,13].

### 2.2. Maximum Potential Typhoon

The typhoon model used in this study is based on Shimokawa et al. [14]. Shimokawa et al. developed a Typhoon Bogus based on the method of Yoshino et al. [15] under the meteorological conditions if global warming proceeds as per the IPCC (Intergovernmental Panel on Climate Change) A1B scenario [16]. The A1B scenario assumes high economic growth and globalization, and a balanced use of fossil fuels and renewable energy. Next, 50 initial meteorological field cases were created by embedding these typhoons at various locations on Japan's southern sea, and storm surge analyses were carried out by using an atmosphere-ocean-wave coupled model. In this study, we define the typhoon which records the maximum storm surge deviation at the Nagoya Port in the study by Shimokawa et al. [14] as the maximum potential typhoon.

The time variation of central pressure and maximum wind speed of the maximum potential typhoon are shown in Figure 1. The central pressure and maximum wind speed at the time of the nearest approach to the Nagoya Port exceeded the central pressure and maximum wind speed of Typhoon Vera, which caused the maximum storm surge at the Nagoya Port. Therefore, the maximum potential typhoon would cause a storm surge greater than Typhoon Vera.

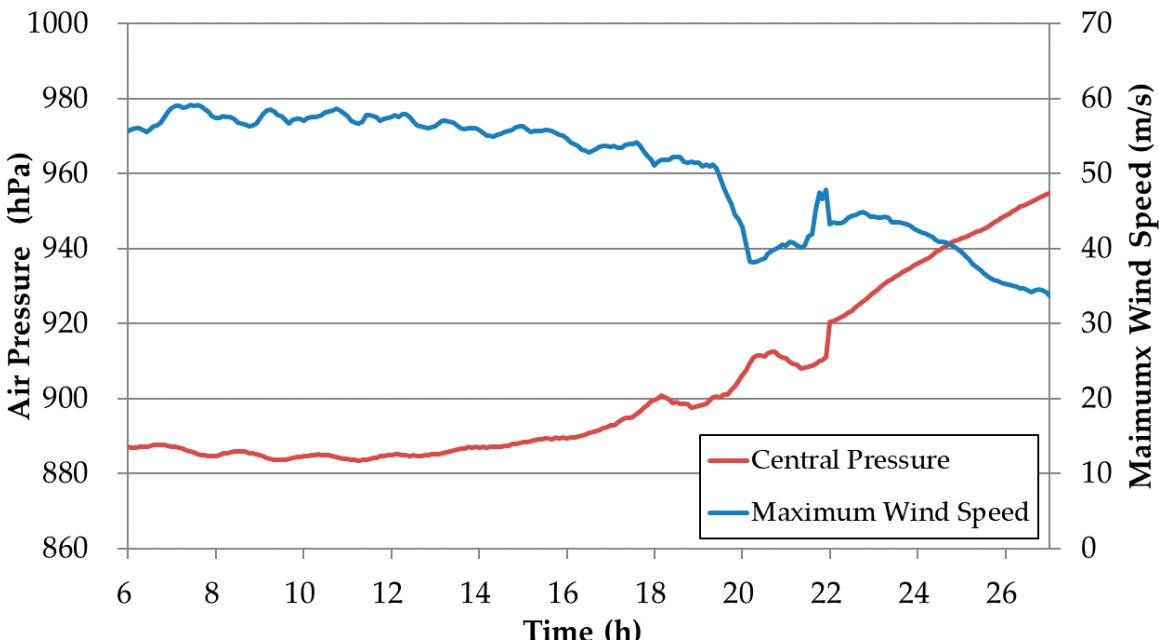

**Figure 1.** Time variation of central pressure (**red line**) and maximum wind speed (**blue line**) of the maximum potential typhoon. The time of *x* axis is matched to the analysis time.

The spatial distribution of air pressure and wind speed of the maximum potential typhoon at *t* = 19:00 are shown in Figure 2. This is the time when the typhoon approached the Nagoya Port. The wind direction at the mouth of Ise Bay is south, so the seawater enters the Ise Bay. The air pressure and wind field of the maximum potential typhoon are created in one-hour increments, and grid sizes are 810 m. In storm surge analysis, the air pressure and wind field are linearly interpolated in the temporal and spatial directions.

### 2.3. Calculation Model

In this study, tsunami simulator T-STOC (Tsunami-Storm surge and Tsunami simulator in Oceans and Coastal areas), developed by PARI (Port and Airport Research Institute) is used [17]. T-STOC consists of a hydrostatic model (STOC-ML) and a non-hydrostatic model (STOC-IC). Those two models can be connected by nesting method. At the nesting boundary, the parent and child domain communicate the water level and velocity of mesh each other. The ratio of the grid size is usually set at 3:1 or 5:1. In this study, we applicate STOC-ML to all domains. The detail of STOM-ML is described in Reference [17,18].

Since STOC-ML is a hydrostatic model, the pressure is expressed as a function of the distance from the water surface. On the other hand, the wind field is used to calculate the wind stress.

The high reproducibility of water level and velocity of T-STOC has been verified [17–19]. The application of T-STOC to storm surge analysis was verified by the authors [20].

**(a) Air Pressure**

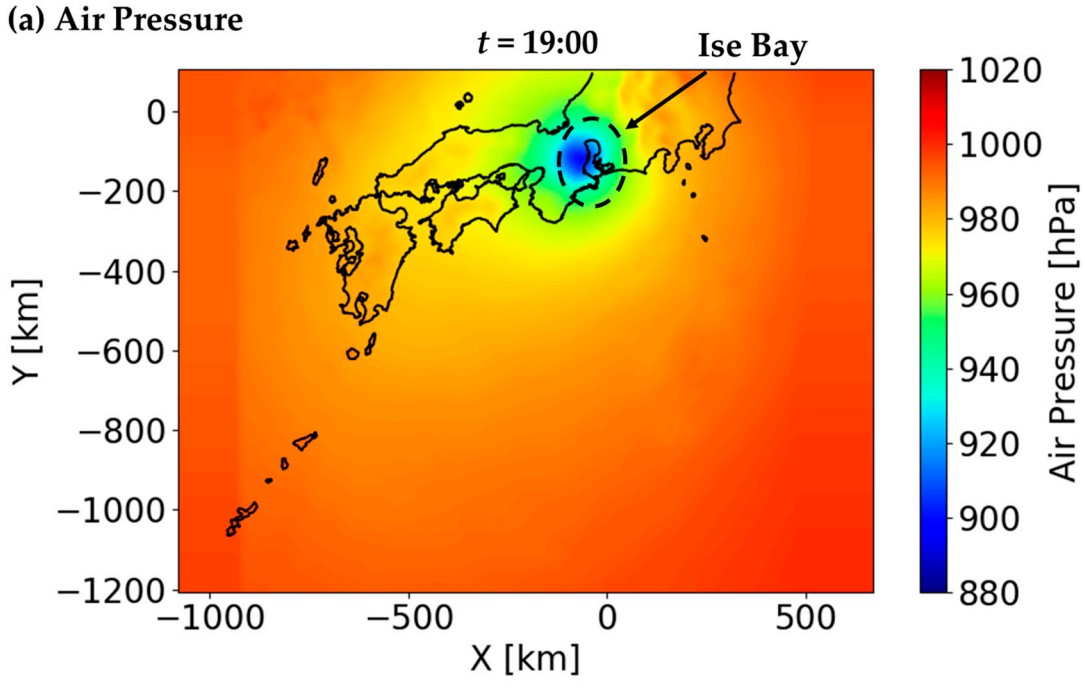

**(b) Wind Speed and Wind Direction**

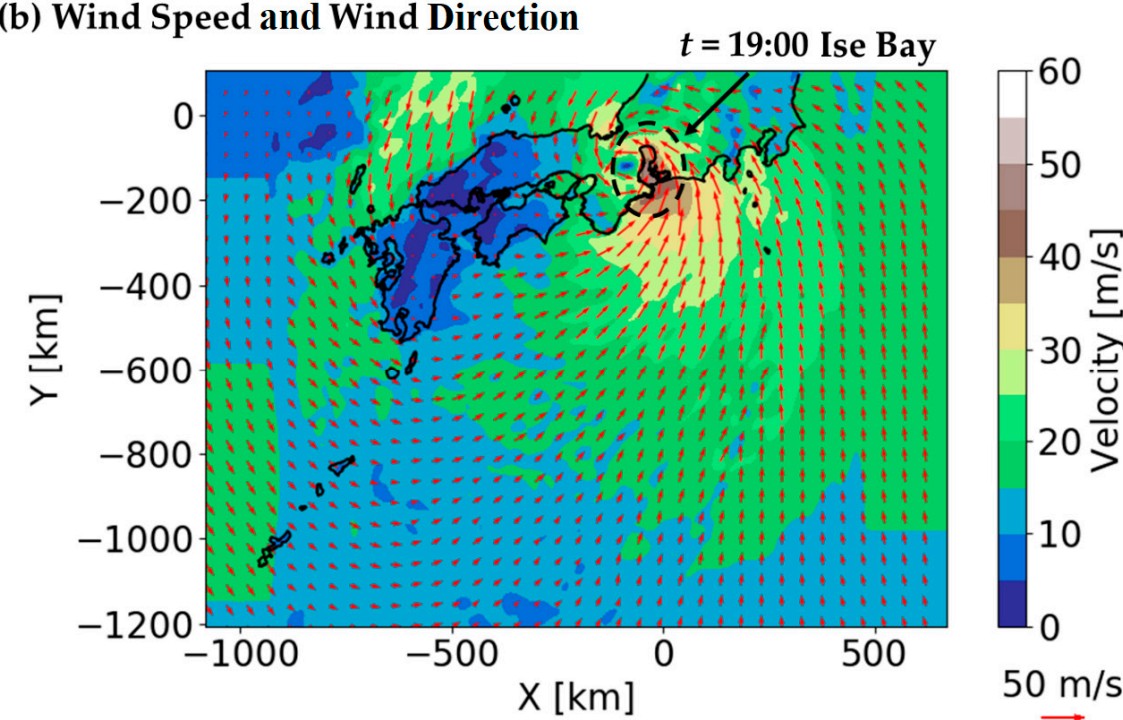

**Figure 2.** Spatial distribution of central pressure and maximum wind speed of the maximum potential typhoon at *t* = 19:00. (**a**) Air Pressure. (**b**) Wind Speed and Wind Direction.

*2.4. Geometry Model*

The analysis domains are shown in Figure 3. The analysis domains are six domains (Dom 1 to Dom 6) connected by a grid nesting method. Each domain's mesh sizes are 810 m, 270 m, 90 m, 30 m, 10 m, and 2 m. In the analysis domains, Dom 3 to Dom 6 are defined as inundation domains. We constructed a geometry model for Dom 1 to 5 based on the sea depth and the ground height

published by the Cabinet Office of Japan [21]. The analysis mesh's vertical direction in all domains was divided into two layers at a depth of 10 m to consider the wind-driven effect and the computing time.

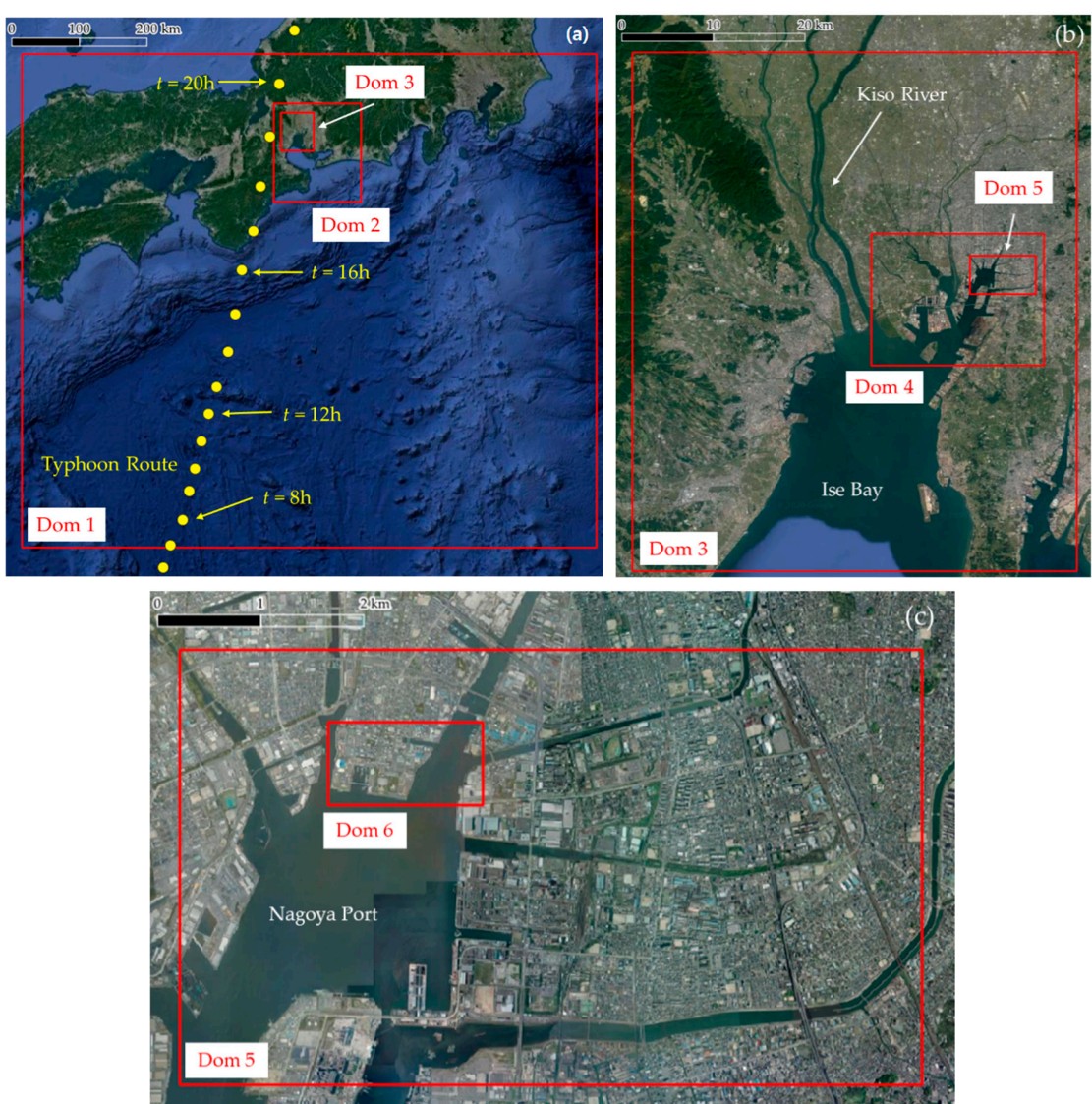

**Figure 3.** Analysis domains (red boxes) in this study. (**a**) Dom 1 (dx = 810 m) to Dom 3 (dx = 90 m). (**b**) Dom 3 to Dom 5 (dx = 10 m). (**c**) Dom 5 to Dom 6 (dx = 2 m). Yellow circles in (**a**) show the typhoon route. The background photos are taken by the Geospatial Information Authority of Japan.

In addition, we constructed a geometry model considering the shape and layout of buildings for Dom 6 (Figure 3). The domain size is East–West 750 m by North–South 400 m. The building model was created from the results of aerial laser surveying with a resolution of 2 m. The ground height in Dom 6 is based on Dom 5 ground height and reflects the height of buildings.

The boundary conditions of the bottom surface in Dom 1 to Dom 5 are Manning's n roughness. In Dom 6, several methods for setting roughness at the urban spatial scale have been proposed for tsunami [8,22], but the standard method is not clear. In this study, we considered that the flow inhibition is represented by the shape and layout of buildings, so Manning's n roughness is not used. And the boundary conditions of the bottom surface and the wall surface of buildings in Dom 6 are No-Slip condition.

Figure 4 shows the seawalls, and the seawall doors are placed in black line and yellow line as linear boundary grids. The seawall doors are called in order from the southwest side, Door A to Door

G. P1 and P2 are the output points for the time series of water level and are described in detail in Section 4.2. The height of the seawalls was set to T.P. + 4.5 m (T.P. means Tokyo Peil) based on the field survey. A field photo of Door C is shown in Figure 5. The photo was taken from inside the seawall. The picture shows that the road is passable, but, during a typhoon, the road is closed by the seawall door.

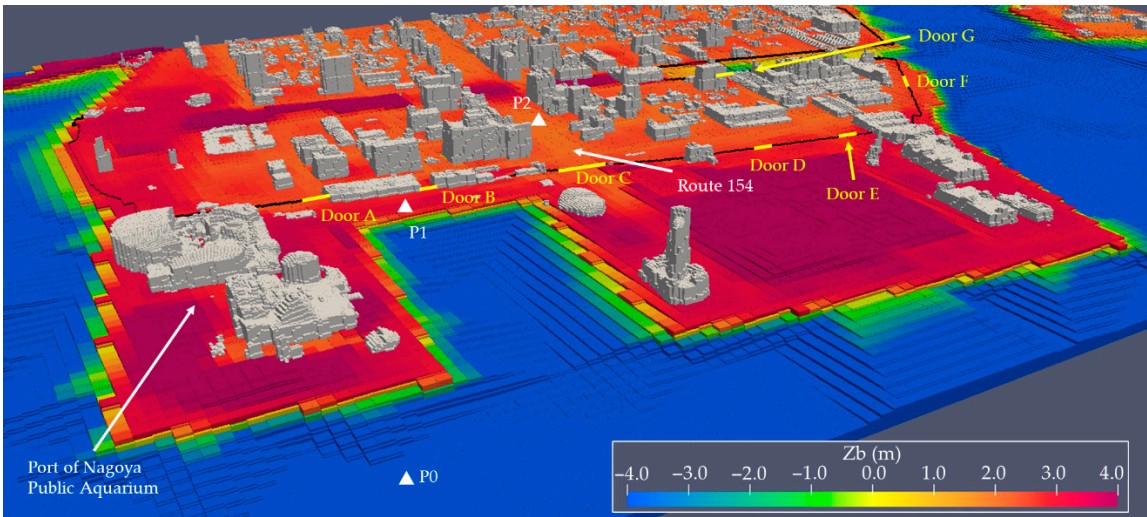

**Figure 4.** Geometry model of Dom 6. Color bar is ground level, gray polygons are building shape, black lines are seawall, and yellow lines are seawall doors. The inundation analyses were carried out with the seawall doors in opened and closed conditions. White triangles (P0, P1, and P2) are the output points of time variation data.

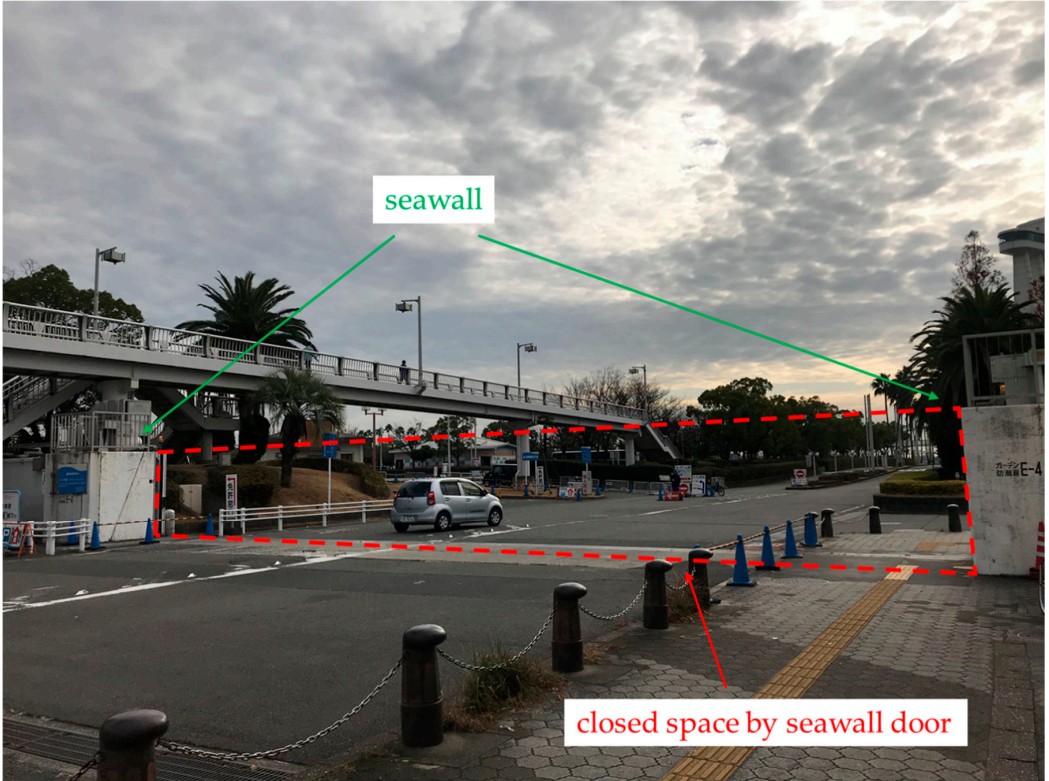

**Figure 5.** A field photo of Door C from inside the seawall. When a typhoon strikes, a space between the seawall (red dashed line) is closed by the seawall door.

Although there are several subway stations in this study area, we do not consider the inflow into subway stations because the entrances to subway stations are protected from flooding by tide protection doors during a typhoon. However, the intrusion of seawater into the subway station by the destruction of the seawall door is important as disaster prevention measures, so it needs to be considered in the future.

### 2.5. Calculation Condition

Using the first 6 h of analysis, the air pressure and wind field of typhoon are developed to *t* = 6:00 in Figure 1 at the same location. The analysis time of moving the typhoon is 21 h (*t* = 6:00–27:00 in Figure 1), and the total time is 27 h. The delta time was varied to satisfy the CFL (Courant Friedrichs Lewy) condition.

The tidal conditions were fixed at T.P. + 1.68 m, which is the sum of the syzygy average high tide level at the Nagoya Port (T.P. + 1.20 m) and the maximum sea-level rise predicted by the IPCC A1B scenario (0.48 m). The syzygy average high tide level means the average of high tide level observed within 2 days before and 4 days after the full and new moons of each month. In order to consider the worst conditions where the inundation area will be maximum, the tide level was set as a fixed condition. The initial water level of rivers is equal to the tide level, and we do not take into account the inflow from the river.

Analysis cases are two cases, one with all the seawall doors opened and the other with all the doors closed. The operational rules of seawall doors are to be closed at storm surge. There are two reasons for the comparison between the opened and the closed conditions. First, we examine the effect of closing the seawall doors on disaster mitigation. Second, we consider the possibility that we may not be able to close the seawall doors due to trouble.

## 3. Results

### 3.1. Storm Surge and Inundation Areas of Inner Part of Ise Bay

The time variation of water level at P0 (shown in Figure 4) is shown in Figure 6. The case is the closed condition, but the water level is almost the same, even in the opened condition. The maximum water level is T.P. + 6.07 m, which is 2.18 m higher than the maximum water level observed during Typhoon Vera.

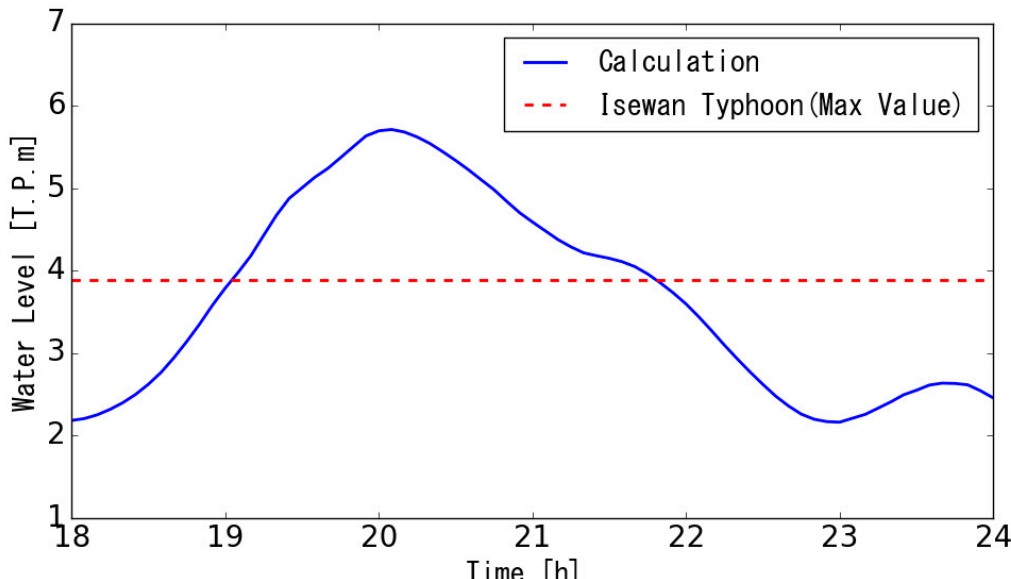

**Figure 6.** Time variation of water level at P0 (shown in Figure 4). The blue line shows the calculation result, and the red dashed line shows the maximum water level observed during Typhoon Vera.

Next, the spatial distribution of the maximum inundation depth in the inner part of Ise Bay is shown in Figure 7. In the inner part of Ise Bay, there are large areas where the elevation is lower than the sea level. When a storm surge greater than the tidal protection facilities occurs, large areas are inundated. The tidal protection facilities are designed to protect the storm surge by Typhoon Vera. However, as shown in Figure 6, the storm surge by the maximum potential typhoon is higher than that by Typhoon Vera. As a result, inundation areas spread around the Nikko, Shohnai, and Tenpaku river basins, where the elevation is lower than the sea level. The area and depth of inundation did not differ much between seawall doors opened and closed conditions.

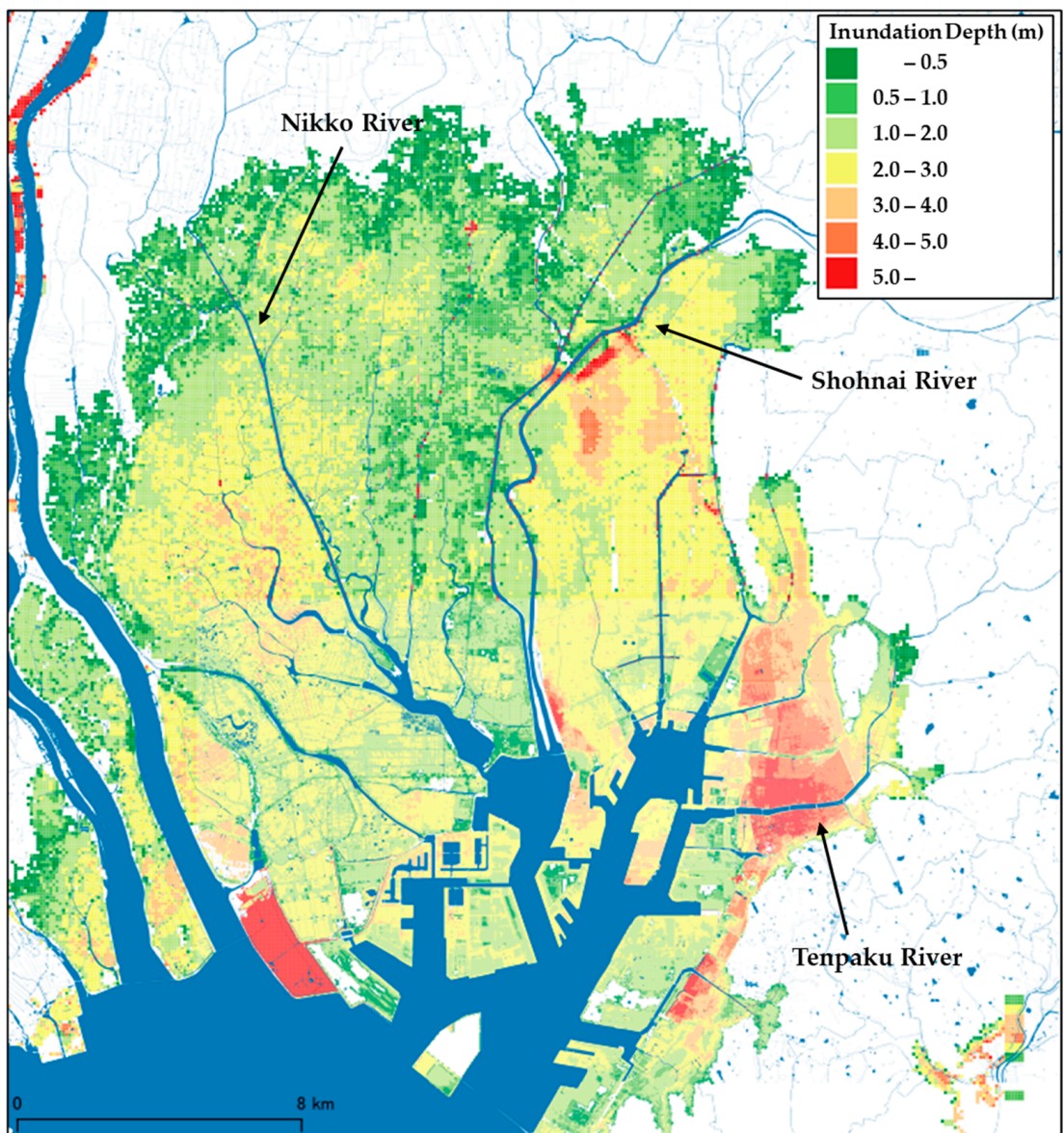

**Figure 7.** Spatial distribution of the maximum inundation depth in the inner part of Ise Bay by the maximum potential typhoon.

*3.2. Inundation Process in Geometry Model Considering Building Shape and Layout*

We carried out storm surge inundation analyses under opened and closed conditions of seawall doors. As an example of the analysis results, the spatial distribution of velocity under seawall doors opened conditions is shown in Figure 8. Figure 8 shows the view of the Dom 6 from the southeast. At *t* = 18:52 (Figure 8a), the storm surge, which entered through Door C, spreads to the left and right

and heads north on Route 154, and the part of the flow is divided by the building layout. On the other hand, the storm surge entering through Door G on the northeast side limits the flow path to the road between buildings, and the velocity of flow is about 4 m/s. This flow head south and then turns westward under the influence of the ground level. The flow field inside the seawall is complex due to the influence of buildings and microtopography.

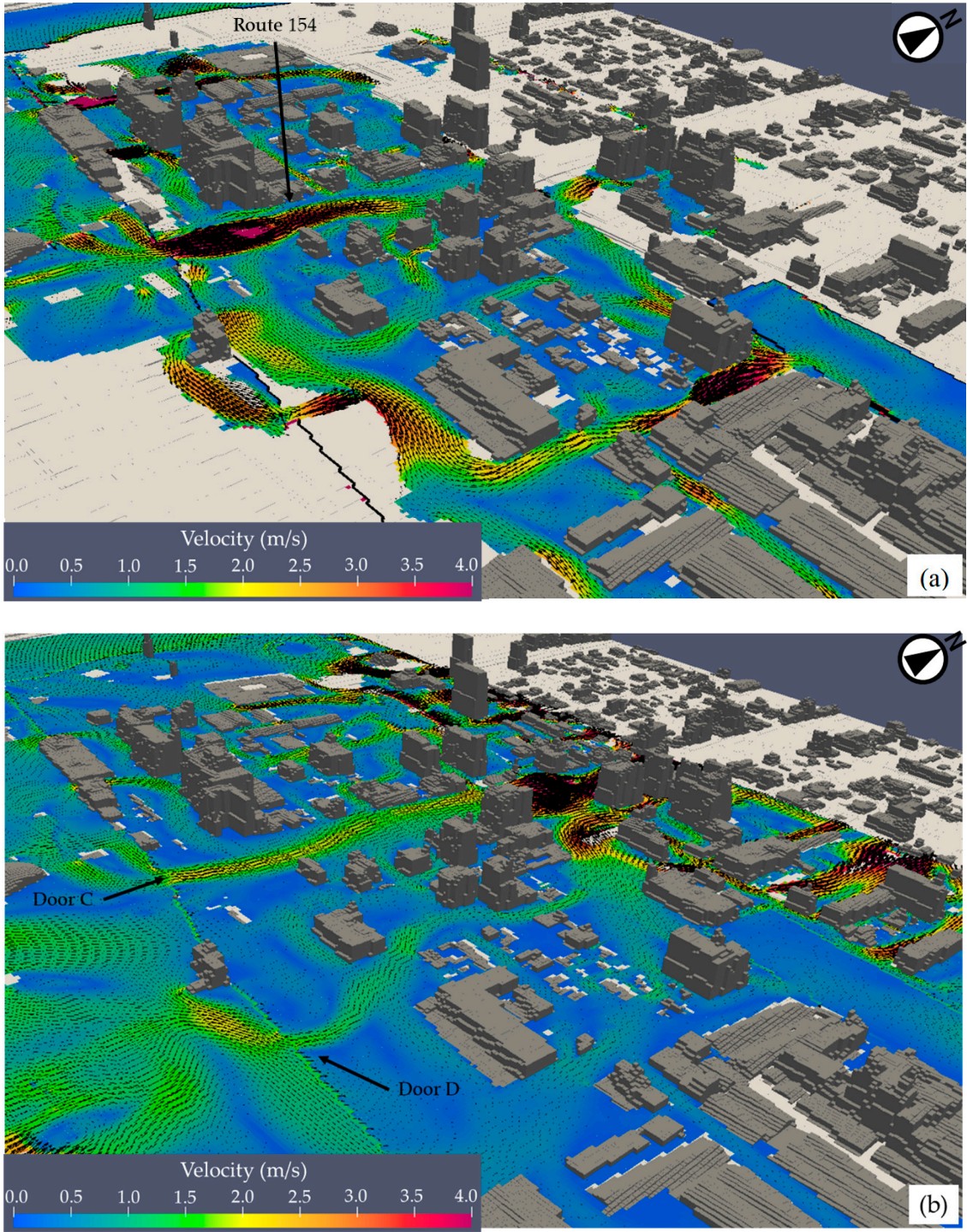

**Figure 8.** Spatial distribution of the velocity vector around the door C. The seawall doors are opened. (**a**) The spatial distribution of *t* = 18:52, when the water level is lower than the seawall height. (**b**) The spatial distribution of *t* = 19:12, when the water level is higher than the seawall height.

The spatial distribution of velocity at *t* = 19:12 is shown in Figure 8b. Almost the entire area inside the seawall levee is inundated because the water level is higher than the height of the seawall. The storm surge entering through Door C and Door D flows northward, but the velocity is lower than *t* = 18:52 due to the increased depth of inundating.

Next, an example of the spatial distribution of velocity at the same time under seawall doors closed condition is shown in Figure 9. In the case of *t* = 18:52 (Figure 9a), the land inside the seawall shows no inundation due to the protective effect of the seawall and seawall doors. At the same time, large areas outside the seawall are inundated, and the flow heads eastward in front of the seawall.

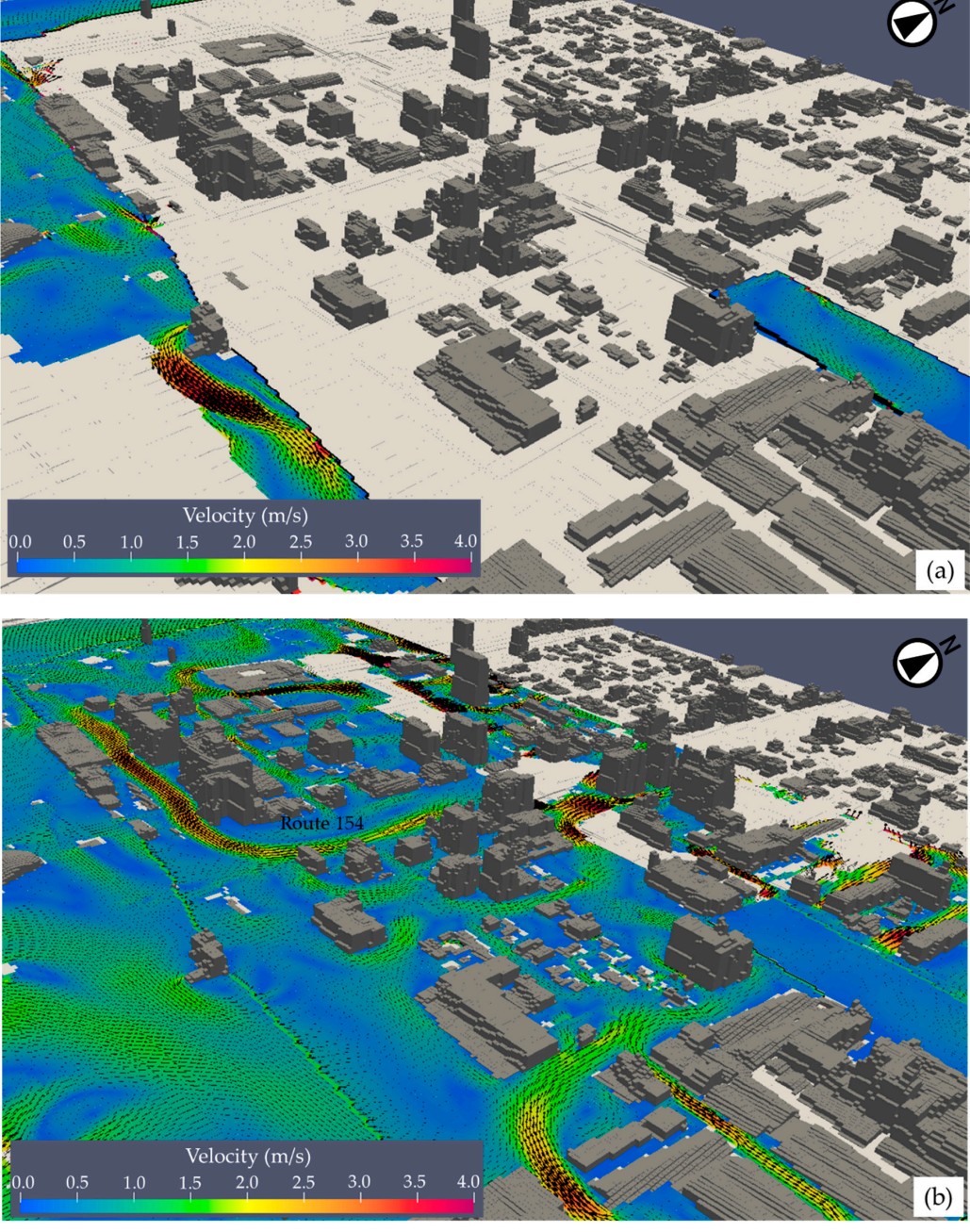

**Figure 9.** Spatial distribution of the velocity vector around the door C. The seawall doors are closed. (**a**) The spatial distribution of *t* = 18:52, when the water level is lower than the seawall height. (**b**) The spatial distribution of *t* = 19:12, when the water level is higher than the seawall height.

On the other hand, at $t$ = 19:12 (Figure 9b), the land inside the seawall is inundated by the seawall's overflow, and the inundation area is almost identical to the area under the seawall doors opened condition. However, the flow field is different. In this case, the eastward flow that passes between the buildings to Route 154 dominates.

By the way, we do not consider changing the seawall height in this study. We would like to consider raising the seawall because it is useful as a protection measure against storm surges.

## 4. Discussion

### 4.1. Comparison of Maximum Velocity Distribution

The spatial distribution of maximum velocity is shown in Figure 10. The sea area is shown in white color, and buildings that are not inundated are gray. In the seawall doors opened condition (Figure 10a), the maximum velocity is more than 10 m/s in the narrow streets between buildings. The width of the road in Dom 6 is narrower than 10 m, except for Route 154. Therefore, this is the effect of the high-resolution geometry model considering the shape and layout of the buildings. In the seawall doors closed condition (Figure 10b), the flow velocity in the narrow streets between buildings is also fast, and the area where the flow velocity is faster than 10 m/s is larger than the opened condition. In such a fast flow, damage to the seawall and the seawall doors may lead to increased inundation.

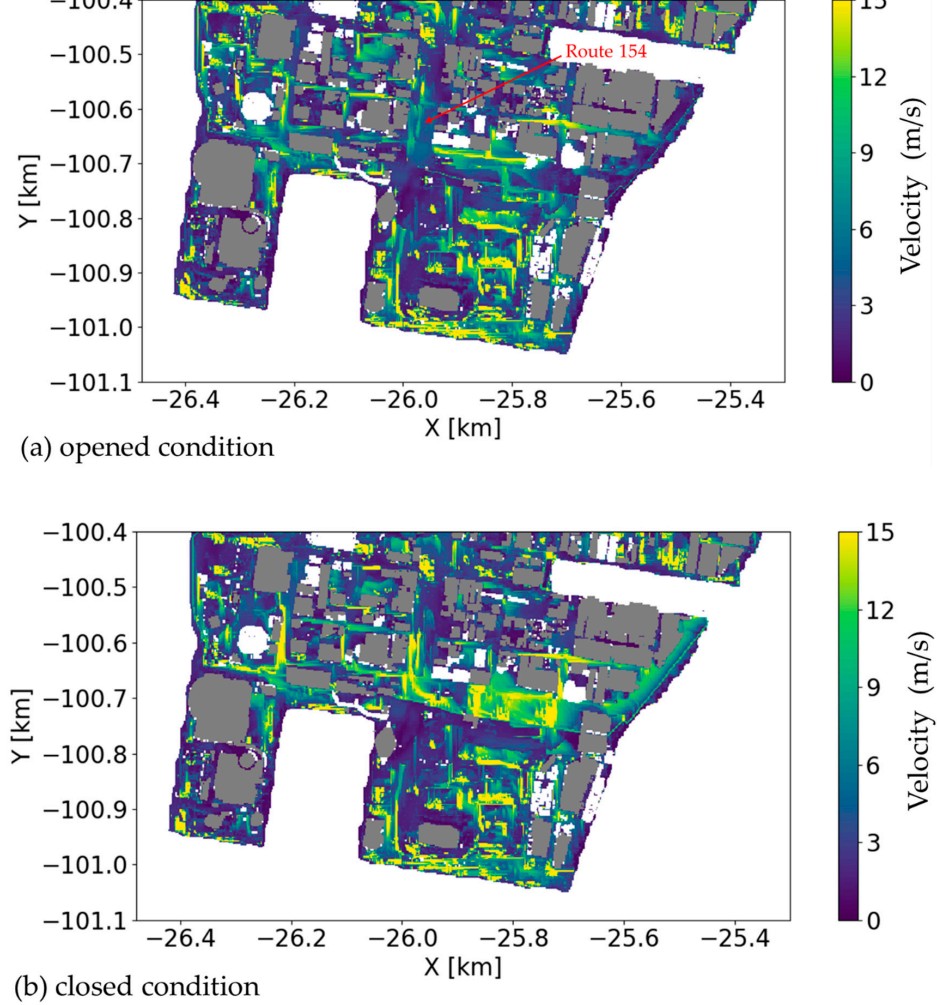

**Figure 10.** Spatial distribution of the maximum velocity in Dom 6. (**a**) Opened condition. (**b**) Closed condition.

The spatial distribution of maximum velocity difference is shown in Figure 11. The figure shows the velocity difference between opened condition (Figure 10a), and closed condition (Figure 10b), and the areas with higher velocity under opened condition indicate positive, while the areas with lower velocity indicate negative. There is no significant difference between the two cases on the outside of the seawall, except for the front of the seawall door D and Door E. On the other hand, the velocity difference in most areas is negative on the inside of the seawall, so the maximum velocity is higher under the closed condition. For this reason, the maximum velocity is mainly recorded at the time when the storm surge overflows the seawall in the closed condition. At that time, the land inside the seawall has been already inundated under the opened condition, so the maximum velocity under the opened condition is probably suppressed because the inundation depth is deeper than the closed condition.

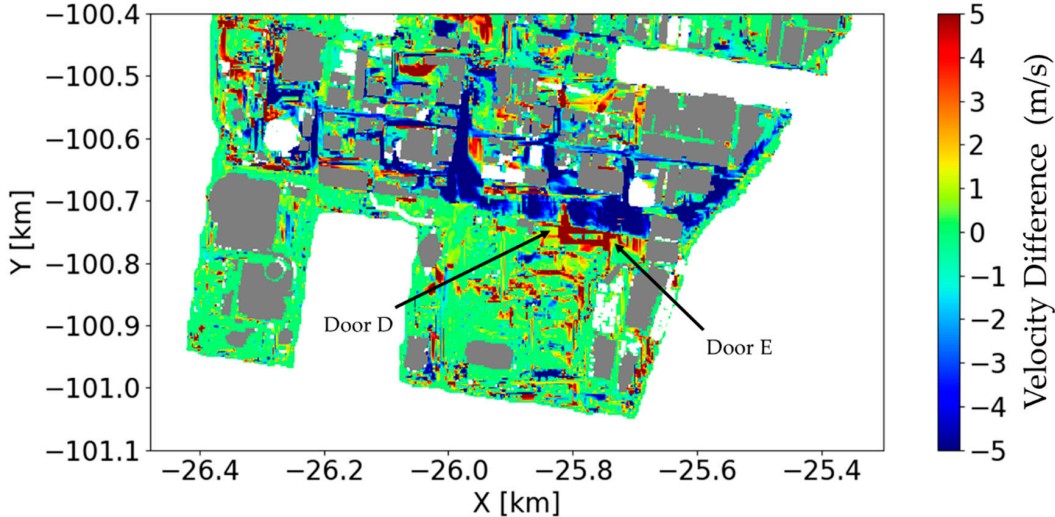

**Figure 11.** Spatial distribution of the maximum velocity difference (opened-closed) in Dom 6. The areas with higher velocity under opened condition indicate positive, while the areas with lower velocity indicate negative.

*4.2. Effects of Seawall Doors Opened/Closed Condition on the Inundation Speed*

The time variation of the water level at P1 and P2 (shown as white triangles in Figure 4) is shown in Figure 12. The value of the graph before the waveform start to rise means the ground level. At P1 (Figure 12a), which is outside the seawall, there is not much difference in water level trend, except for the rise speed of water level is slightly higher under the closed condition. On the other hand, at P2 (Figure 12b), inside the seawall, the start time of water level rise is later under opened condition. It can be said that this is a delay effect of the inundation start time by the seawall doors closed.

On the other hand, the rise speed of water level after inundation is faster under the closed condition. The water level under the closed condition rise rapidly to about T.P. +5.0 m, so the evacuation after inundation started is more difficult than the opened condition. The inundation inside the seawall occurs as the storm surge exceeds the seawall, so it is not easy to predict the inundation start area and the inundation process under the closed conditions. Therefore, the evacuation after inundation under the seawall doors closed condition is more difficult than opened condition. Furthermore, there is almost no difference in the maximum inundation depth between the opened and closed conditions. Therefore, when the storm surge is higher than the seawall, the seawall's effect on disaster mitigation is considered to be small.

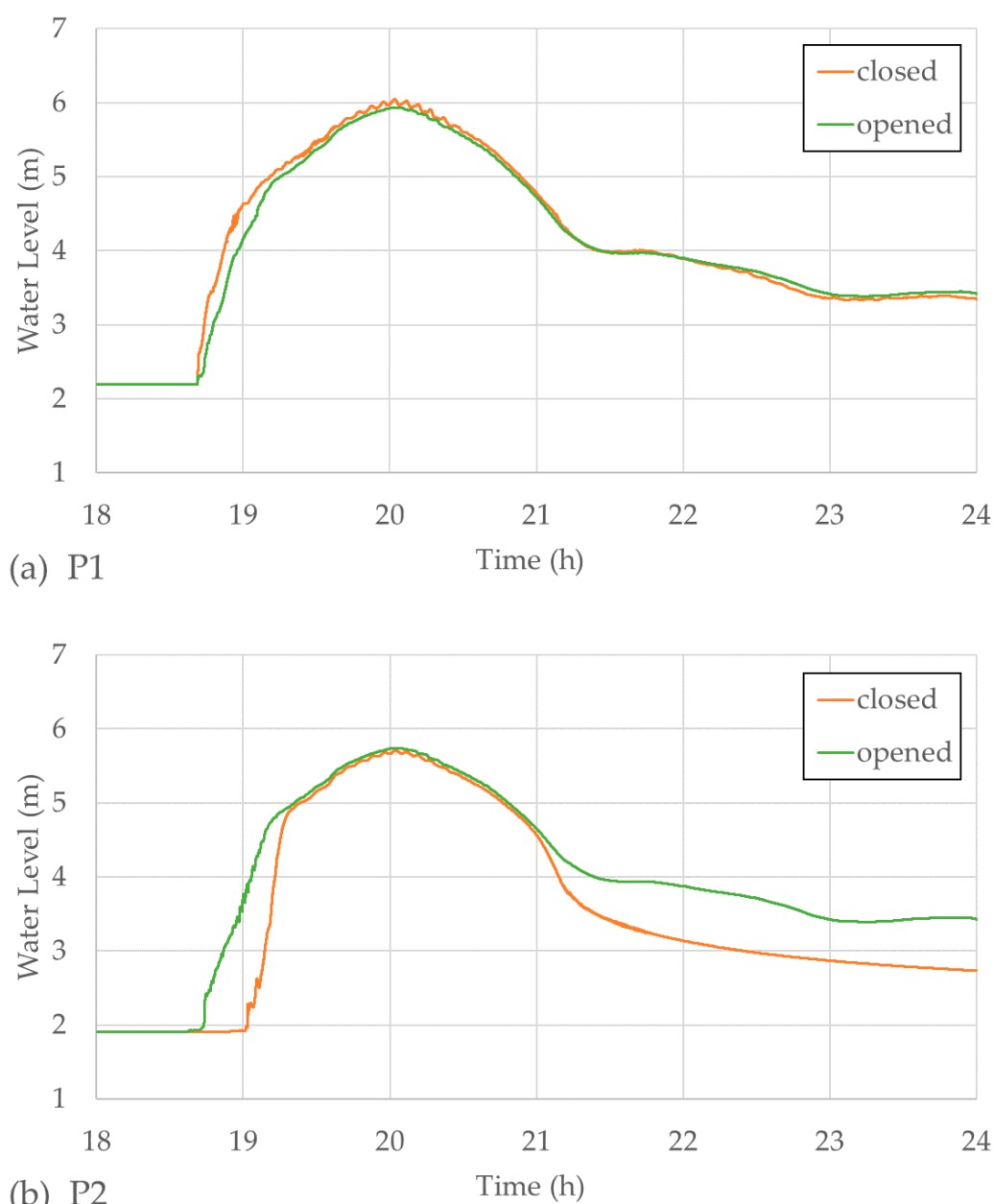

**Figure 12.** Time variation of water level (**a**) P1 and (**b**) P2 (shown as white triangles in Figure 4).

## 5. Conclusions

In this study, storm surge inundation analysis by maximum potential typhoon was carried out around the Nagoya Port area in the inner part of Ise Bay. We constructed a high-resolution geometry model considering the shape and layout of buildings and investigated the storm surge inundation process in detail. The analysis results show that the highest water level by the maximum potential typhoon was about 2 m higher than that by Typhoon Vera, which caused major flooding damage to the area in the past. As a result, the storm surge overtopped the seawall and inundated large areas.

When the areas inside the seawall are inundated by storm surge, the inundation process is strongly influenced by the buildings and becomes complex. The flow inside the seawall is concentrated on the roads between the buildings, and its velocity is high. Some of the roads have maximum flow velocities exceeding 10 m/s under the seawall doors closed condition.

If the storm surge overtopped the seawall, the rise speed of water level inside the seawall is fast, and the evacuation after inundation started is difficult. If the storm surge as high as in this study

would occur, then it is more important to mitigate the personal suffering through hazard maps and warning in advance than to protect against inundation.

**Author Contributions:** Methodology, M.N., S.N., K.K., T.M., and S.S.; investigation, M.N., S.N., K.K., T.M., and S.S.; resources, T.M. and S.S.; writing—original draft preparation, M.N.; writing—review and editing, S.N. and K.K.; supervision, K.K.; funding acquisition, T.M. and S.S. All authors have read and agreed to the published version of the manuscript.

**Funding:** This work was supported by JSPS KAKENHI Grant Number JP18K04377.

**Conflicts of Interest:** The authors declare no conflict of interest.

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
