# Peer review of "Storm Surge Inundation Analysis with Consideration of Building Shape and Layout at Ise Bay by Maximum Potential Typhoon"

_jmse, doi:10.3390/jmse8121024_

Round 1
Reviewer 1 Report
Interesting research on a very relevant topic. Quality of the underlying computations look very good. However, the structure of the paper is somewhat confusing, and the conclusions are not very to-the-point.
Basically the research question of the paper is to investigate what is the additional risk of open/closed flood-doors in a tsunami during situations with waterlevels higher than the design flood. Unfortunately this research question is not formulated in this way in the paper, the reader has to find out him/herself.
The assumption in the paper is that during a hydraulic load higher than the design load the urban area of Ise Bay will be flooded due to overflow of the tsunami wall. An important (political) management question if the damage (expressed in money, human lives, loss of cultural values, etc) will be larger when the flood doors are open or closed during such an extreme event.
The presented model gives values on which such analysis can be made. However, this is not addressed in the paper. This is a pity, it decreases the final value of the paper.
Another use of the described model is the determination of the optimal design height of the flood wall. With a low flood wall the area inside the protected are will be damaged by floods frequently, but the damage is rather limited. The higher one makes the flood walls, the more infrequent damage will be. But in the (rare) occasion of overflow of the flood wall the inundation will be deep and sudden. Also because it very seldomly happens, inhabitants are not prepared.
Not mentioned in the paper is the probability of the Typhoon Vera, neither the probability of the used extreme typhoon (the term “maximum potential typhoon” is confusing, there is no maximum; although with very small probability, there is no real physical limit to a typhoon).
For the computation the STOC-ML model is used. This model has been explained elsewhere. Mentioning the basic equations 1- 5 used by the model are therefore not so relevant in this paper, especially not without detailed explanation. I suggest a good reference to STOC-ML (maybe [14] or [15] ) and omit the equations.
A weak point in the calculation is the calibration of the flooded urban detailed model (Dom6). For the other, larger domains the calibration if this model is not an issue, there is ample experience. But roughness of narrow streets, buildings and other objects in a city is quite unknown, and there is not much prototype experience with this. The same is regarding the extra storage of water flowing into the buildings. In areas like this there are quite large basements which can be filled with water. Is this relevant? See for example Flood-risk assessment of the dense downtown in Fukuoka City, Japan by H. Hashimoto & Y. Nonaka from 2012 (but there are many more papers on this topic).
The paper shows clearly the effect of the overflow of the wall, as well as the high velocities in the street. It is therefore a pity that the damage due to the high velocities are not mentioned. I fear that when flood currents in the order of 10 m/s occur in the opening of the flood door, most probably the foundation of the doors may be washed away. Also when the wall is overtopped, considerable damage will occur at the inner side of the wall (see attached photo) and there is a significant risk that the wall will collapse. It might be good to mention this.

Reviewer 2 Report
The authors described the effects of very high water levels caused by an extreme typhoon. Water levels within an urban area are calculated with a numerical model and the influence of the closing or opening of the sea wall doors is thoroughly discussed.
General Remarks: It would be nice if you could add some information about your area of interest for readers, which are not familiar with Japan. E.g. in line 34 you mention a maximum tidal deviation. Could you please also add some information about the astronomical tides and the tidal range in this area? Please define, what you mean when you talk about a storm surge and its heights (compared to mean sea level, mean high water,…..) - see also remark for line 33.
You changed the order of the pictures for open and closed doors between figures 6-7 and figure 8- it would be easier to follow your thoughts if you would always first explain the cases with open doors and then with closed doors.
You discuss the influence of the open or closed sea wall doors. Nevertheless, the motivation for this discussion is not clear. If a typhoon is approaching your area and you can’t forecast yet, whether the surges will be higher than the sea walls, then the doors will always be closed – won’t they? In which case will you leave the doors open?
Remarks about details in the text:
Abstract:
- inner part of Ise Bay – please add the larger area (coast of Japan…..)
- but the time to spare for evacuation after the onset of inundation is shortened when the high tide level exceeds the seawall à time is shortened compared to what? Not closing the door? Please specify
Line 33: maximum tidal deviation of 3.55m --> compared to what? Mean sea level? Mean high tide? Please specify
Line 59: please discuss, why this is the maximum potential typhoon. What happened, if you choose another Climate Scenario? Or is it the maximum typhoon calculated in Shimokawa et al.?
Line 65: storm surge deviation -> deviation from what? Mean high water?
Figure 1: typo in the legend (Centran pressure instead of Central pressure)
Line 70: ...which is recorded as the maximum storm surge tide -->please rephrase- a storm is not equal to a storm surge – it may cause one.
Line 70: Therefore, the maximum potential typhoon would occur storm surge greater than Typhoon Vera --> please rephrase: it would cause,….
Line 77 and further: Calculation model: You wrote down the (well known) equations of motion, but you did not explain the boundary conditions, which influence the solution, such as e.g.
- How did you prescribe the outflow of the rivers? Water levels before the typhoon?
- Wind field: horizontal grid sizes? Time steps? Wind directions?
Please specify.
Line 130: syzygy average high tide level: specify the difference to high tide level.
Line 130: please define T.P.
Line 164: north on Route 154 – please show the route in the map
Conclusions line 268: Because the maximum potential typhoon used in this study is more powerful than Typhoon Vera, so it is more important to mitigate the personal suffering through hazard maps and warning in advance than to protect against inundation. -->This sentence could be misunderstood. If the water levels are not as high as your walls, the protection against inundation makes a big difference for the safety of the people! Please rephrase it. Perhaps: if the maximum surge would be as high as in our study, then…..
Reviewer 3 Report
This is an interesting study on the storm surge flooding under global warming scenario and considering mitigative structures (seawalls). Building footprints are accounted for in terms of flooding extent and velocity. The 3-D representations in details are impressive. Overall I don't have major concerns. One thing missing from the paper is the discussion on bottom friction (see point 5):
1. Line 130: Why not use the tidal time series? What's the benefit of using constant tidal boundary?
2. I assume T.P. is Tokyo Peil, maybe it's better to include this definition at the beginning.
3. Since there was no velocity measurement, I assume the authors trust the model results to be very close to reality, but it'd be good to mention the uncertainty.
4. Line 244: what is the margin of time for evacuation? Any definition on this?
5. Can you also describe bottom friction/roughness set up in your model? During overland flood, bottom friction might be a dominant factor.
Reviewer 4 Report
The authors are dealing with a very interesting subject i.e. coastal inundation due to storm surge. They are exploring the inundation in building areas in order to examine different protecting conditions to avoid civil damages.
But their manuscript is very sort without presenting or giving information about the main scientific questions rising through the paper.
Following are some specific comments to consider for the revision of the manuscript.
A section should be added (possibly before 2.1) in order to give more information about the area, and the procedure followed in this research and about the typhoon events is the area studied.
Line 33 {… in bays with the mouth facing south…}: this in not trough as a general comment. Should be rephrased.
Lines 67-68 {The time variation of central pressure and maximum wind speed of the maximum potential typhoon are shown in Figure 1}: no data given for the typhoon speed.
Line 93 {On the other hand, the wind field is used to calculate the wind stress}: more data should be given.
Line 98 {a grid nesting method}: more data should be given about the nesting method.
Lines 100-102 {The analysis mesh's vertical direction was divided into two layers at a depth of 10 m to consider the wind-driven effect and the computing time}: Is this valid to all sub-domains or only to sub-domains 1 to 3?
Lines 111-112 {In addition, we constructed a geometry model considering the shape and layout of buildings for Dom 6 (Figure 3)}: This means that in other sub-domains the model depth is following the sea-bottom and the ground-height (altimetry) without buildings. Is that correct?
Lines 114-115 {The boundary condition of buildings is set to No-Slip condition}: No information are given for the boundary conditions in the rest area (the area without buildings).
Line 117 {T.P. + 4.5 m}: T.P.=?
Line 117 {based on the field survey}: no data are given.
Lines 127-128 {The analysis time was 27 hours, and the first 6 hours were used to develop the typhoon at the same location}: The procedure followed for the typhoon input data to the simulation process it is not clear. In fig.2, according to time presented on the typhoon route, the typhoon is entering Dom 1 at t=7h and exit at t=~20h.
Fig.5: The max. inundation depths in the Dom 4.
Line 159: fonts!
Line 163 {t = 18:52 (Figure 6(a))}: Is this time chosen for a specific reason?
Line 169 {t = 19:12}: Is this time chosen for a specific reason?
Lines 176-178 {In some parts of the seawall, the current velocity increases with the overflow. In some parts of the seawall, the overflow velocity is high.}: These comments are very general and are not in agreement with specific areas of the fig.7.
Lines 236-237 {The time before the inundating is shown as the ground level instead of the water level}: This statement is confusing.
Lines 239-240 {at P2 (Figure 10(b)), inside the seawall, the start time of water level rise is later under opened condition}: This is not the case according to fig. 10b. The water level starts rising earlier under the closed conditions rather the open conditions.

Round 2
Reviewer 1 Report
I found that the authors agreed with my comments 3 and 7, but did not mention them in the revised paper. In my opinion the usability of the paper increases considerably by adding these points into the text of the paper. (for nr. 3 this is an important application of the model, but not yet addressed in this paper but subject to future research, for nr. 7 subway stations are not included, which also an important step for future development of the model).
I think it is good to show to the readers wat are the future steps in research.
Reviewer 4 Report
I am proposing to authors not delete the lines with the equations of the model used (lines 126-139).
